# Development of Liver Fibrosis Represented by the Fibrosis-4 Index Is a Specific Risk Factor for Tubular Injury in Individuals with Type 2 Diabetes

**DOI:** 10.3390/biomedicines12081789

**Published:** 2024-08-07

**Authors:** Tomoyo Hara, Takeshi Watanabe, Hiroki Yamagami, Kohsuke Miyataka, Saya Yasui, Takahito Asai, Yousuke Kaneko, Yukari Mitsui, Shiho Masuda, Kiyoe Kurahashi, Toshiki Otoda, Tomoyuki Yuasa, Akio Kuroda, Itsuro Endo, Soichi Honda, Akira Kondo, Munehide Matsuhisa, Ken-ichi Aihara

**Affiliations:** 1Department of Hematology, Endocrinology and Metabolism, Graduate School of Biomedical Sciences, Tokushima University, 3-18-15 Kuramoto-cho, Tokushima 770-8503, Japan; yamagami.hiroki@tokushima-u.ac.jp (H.Y.); saya.y@tokushima-u.ac.jp (S.Y.); asai.takahito@tokushima-u.ac.jp (T.A.); mitsui.yukari@tokushima-u.ac.jp (Y.M.); m-shiho@tokushima-u.ac.jp (S.M.); kurahashi.kiyoe@tokushima-u.ac.jp (K.K.); 2Department of Preventive Medicine, Graduate School of Biomedical Sciences, Tokushima University, 3-18-15 Kuramoto-cho, Tokushima 770-8503, Japan; 3Department of Diabetology and Metabolism, Tokushima Prefectural Central Hospital, 1-10-3 Kuramoto-cho, Tokushima 770-8503, Japan; 4Department of Internal Medicine, Tokushima Prefectural Kaifu Hospital, 266 Sugitani, Nakamura, Mugi-cho, Kaifu-gun, Tokushima 775-0006, Japan; 5Department of Community Medicine and Medical Science, Graduate School of Biomedical Sciences, Tokushima University, 3-18-15 Kuramoto-cho, Tokushima 770-8503, Japan; otoda.toshiki@nihon-u.ac.jp (T.O.); yuasa.tomoyuki@tokushima-u.ac.jp (T.Y.); aihara@tokushima-u.ac.jp (K.-i.A.); 6Diabetes Therapeutics and Research Center, Institute of Advanced Medical Sciences, Tokushima University, 3-18-15 Kuramoto-cho, Tokushima 770-8503, Japan; kurodaakio@tokushima-u.ac.jp (A.K.); matuhisa@tokushima-u.ac.jp (M.M.); 7Department of Bioregulatory Sciences, Graduate School of Biomedical Sciences, Tokushima University, 3-18-15 Kuramoto-cho, Tokushima 770-8503, Japan; endoits@tokushima-u.ac.jp; 8Minami Municipal National Insurance Hospital, 105-1 Tai, Minami-cho, Kaifu-gun, Tokushima 779-2109, Japan; 9Kondo Naika Hospital, 1-6-25 Nishi Shinharma-cho, Tokushima 770-8008, Japan; 10Department of Internal Medicine, Anan Medical Center, 6-1 Kawahara Takarada-cho, Anan 774-0045, Japan

**Keywords:** Fibrosis-4 index, liver fibrosis, type 2 diabetes, diabetic kidney disease, tubular injury, L-FABP

## Abstract

Although hyperglycemia and hypertension are well-known risk factors for glomerular injury in individuals with type 2 diabetes (T2D), specific risk factors for tubular injury remain unclear. We aimed to clarify the differences between risk factors for glomerular injury and risk factors for tubular injury in individuals with T2D. We categorized 1243 subjects into four groups based on urinary biomarkers, including the albumin-to-creatinine ratio (uACR) and L-type fatty acid-binding protein-to-creatinine ratio (uL-ABPCR) as a normal (N) group (uACR < 30 mg/gCr and uL-FABPCR < 5 μg/gCr; n = 637), a glomerular specific injury (G) group (uACR ≥ 30 mg/gCr and uL-FABPCR < 5 μg/gCr; n = 248), a tubular specific injury (T) group (uACR < 30 mg/gCr and uL-FABPCR ≥ 5 μg/gCr; n = 90), and a dual injury (D) group (uACR ≥ 30 mg/gCr and uL-FABPCR ≥ 5 μg/gCr; n = 268). Logistic regression analysis referencing the N group revealed that BMI, current smoking, and hypertension were risk factors for the G group, creatinine (Cr) and Fibrosis-4 (FIB-4) index were risk factors for the T group, and BMI, hypertension, HbA1c, Cr, and duration of diabetes were risk factors for the D group. While hypertension was a distinct specific risk factor for glomerular injury, the FIB-4 index was a specific contributor to the prevalence of tubular injury. On the other hand, the logistic regression analysis revealed that the hepatic steatosis index (HSI) did not show any significant association with the G group, T group, or D group. Taken together, the development of liver fibrosis rather than liver steatosis is an inherent threat relating to tubular injury in individuals with T2D.

## 1. Introduction

Diabetic kidney disease (DKD) is a common microvascular complication and is the major cause of chronic kidney disease (CKD) and end-stage renal disease [1]. The mortality rate for individuals with DKD is 30 times higher than that for individuals without nephropathy [2], thus manifesting a substantial global predicament, imposing considerable societal and economic burdens.

The classical presentation of DKD is characterized by hyperfiltration and albuminuria in the early phases, which have been considered to be a consequence of progression of glomerular injury [3,4]. In recent years, this concept has been increasingly changed, as evidence suggests that a decline in the estimated glomerular filtration rate (eGFR) may also occur independently from the development of albuminuria, and some of these cases are presumed to be attributable to tubular injury [5].

Albuminuria is utilized as a marker for glomerular injury, while L-type fatty acid-binding protein (L-FABP) is used as a marker for tubular injury. L-FABP is a 14 kDa small molecule that is expressed in the cytoplasm of human proximal tubules. It was shown that the gene expression of L-FABP was upregulated by various types of stress that cause tubulointerstitial damage and that urinary excretion of L-FABP was increased [6].

In patients with DKD, it is well known that glomerular injury is significantly influenced by blood glucose and blood pressure management [1]. However, there remains much uncertainty regarding the specific clinical risks associated with tubular injury.

Recently, interest has been shown in the concept of a liver–kidney interaction, which suggests that metabolic dysfunction-associated steatotic liver disease (MASLD) and CKD may mutually exacerbate each other’s progression [7]. MASLD is a hepatic disorder linked to metabolic abnormalities such as obesity and T2D. However, the detailed mechanisms underlying this relationship remain largely obscure and, within the context of MASLD, there is a spectrum of various pathological states, including simple lipodroplet deposition and forms accompanied by fibrotic alterations [8]. There have been reports indicating an association between liver fibrosis and DKD [9]. However, it remains unclear whether this association pertains to the progression of all types of DKD or is specific to certain types, considering that some basic research reports suggest the possibility of a common pathological mechanism underlying both liver fibrosis and renal tubular fibrosis [10].

Based on the above, this study aimed to verify the potential association between DKD, including tubular injury and MASLD and to determine which specific pathological phenotypes or disease stages within the MASLD spectrum are involved in this relationship.

## 2. Materials and Methods

### 2.1. Subjects and Ethics Statement

We consecutively recruited 1243 Japanese individuals (699 males and 544 females) aged over 20 years who were either outpatients or inpatients diagnosed with T2D. The diagnosis of T2D adhered to the criteria outlined by the Expert Committee on the Diagnosis and Classification of Diabetes Mellitus [11]. All participants were recruited from Tokushima University Hospital, Anan Medical Center, Kondo Naika Hospital, and Minami Municipal National Insurance Hospital between July 2017 and April 2024. The exclusion criteria for the participants were as follows: (1) patients with advanced malignancies, (2) patients with secondary diabetes, such as steroid-induced diabetes or pancreatic diabetes, (3) pregnant patients, (4) patients with malnutrition (serum albumin < 3.0 g/dL), and (5) patients undergoing dialysis.

Information on clinical history, physical findings, biochemical examinations of blood samples and urine samples, and medications was obtained from electronic medical records. Current smokers were defined as individuals who had smoked in the last two years. Body mass index was calculated as an obesity index. Hypertension was diagnosed in those who had systolic blood pressure (SBP) ≥ 140 mmHg and/or diastolic blood pressure (DBP) ≥ 90 mmHg or those receiving antihypertensive drugs. Dyslipidemia was diagnosed in those who had low-density lipoprotein cholesterol (LDL-C) levels of ≥140 mg/dL or triglycerides (TGs) of ≥150 mg/dL or a high-density lipoprotein cholesterol (HDL-C) level of less than 40 mg/dL or those receiving hypolipidemic drugs.

Our retrospective observational study was conducted according to the institutional guidelines of each hospital, including Tokushima University Hospital, Anan Medical Center, Kondo Naika Hospital, and Minami Municipal National Insurance Hospital, and was in compliance with the Helsinki Declaration. It was also approved by the Institutional Review Board of each hospital.

### 2.2. Biochemical Analyses

Blood and single spot urine samples were usually used to determine blood cell counts, plasma glucose (PG), HbA1c, and serum biochemical parameters, including LDL-C, TG, HDL-C, uric acid (UA), aspartate aminotransferase (AST), alanine aminotransferase (ALT), creatinine (Cr), albumin, urine albumin-to-creatinine ratio (uACR), and urine liver-type fatty acid-binding protein (L-FABP)-to-creatinine ratio (uL-FABPCR). PG and serum levels of LDL-C, TG, HDL-C, UA, AST, ALT, albumin, and Cr were measured using enzymatic methods. HbA1c was assayed using high-performance liquid chromatography. Urine albumin levels were assayed by performing turbidimetric immunoassays (TIA-ALBG, catalogue No. 20600AMZ01334000, Serotec, Hokkaido, Japan), and uL-FABP levels were assayed by performing chemiluminescent enzyme immunoassays (Lumipulse Presto L-FABP, catalogue No. 298213, Fujirebio, Tokyo, Japan). eGFR was calculated according to the following formula from the Japanese Society of Nephrology: eGFR (mL/[min·1.73 m^2^]) = 194 × serum and creatinine level^−1.094^ × age^−0.287^ (×0.739 if female) [12]. In addition, the Fibrosis-4 (FIB-4) index, as a clinical marker of hepatic fibrosis, was calculated in each subject by using the following formula: FIB-4 index = Age (years) × AST (U/L)/[platelet count (×10^9^/L) × √ALT (U/L)] (<1.3 being categorized as low risk, 1.3 to less than 2.67 being categorized as a high risk of fibrosis, and ≥2.67 being categorized as cirrhosis) [13]. The hepatic steatosis index (HSI), as a clinical marker of fatty liver, was also calculated in each subject by using the following formula: 8 × (ALT/AST ratio) + BMI (+2 if female; +2 if diabetes mellitus) (>36 predicting the presence of fatty liver) [14].

### 2.3. Classification of Participants Based on Urinary DKD Biomarkers

Although the previously reported cutoff value of uL-FABPCR for identifying tubular injury was ≥4.7 μg/gCr [15], in this study, the cutoff value of uL-FABPCR for identifying tubular injury was set to 5 μg/gCr as the highest quartile. The cutoff value of uACR for identifying glomerular injury was set to uACR 30 mg/gCr as the diagnostic criterion for microalbuminuria. We divided the subjects into four groups based on these cutoff points as follows: a normal group (Group N: uACR < 30 mg/gCr and uL-FABPCR < 5 μg/gCr), a glomerular specific injury group (Group G: uACR ≥ 30 mg/gCr and uL-FABPCR < 5 μg/gCr), a tubular specific injury group (Group T: uACR < 30 mg/gCr and uL-FABPCR ≥ 5 μg/gCr), and the dual injury group (Group D: uACR ≥ 30 mg/gCr and uL-FABPCR ≥ 5 μg/gCr).

### 2.4. Statistical Analyses

The Shapiro–Wilk test was used to assess the normality of continuous variables. Continuous variables exhibiting a normal distribution were presented as means ± standard deviation (SD), while those with a non-normal distribution were presented as medians (Q1, Q3). Categorical parameters were expressed as percentages and numerical counts. Males, the presence of hypertension, dyslipidemia, and current smokers were encoded as dummy variables.

Differences in the mean values between groups were tested using the Kruskal–Wallis test and Dunn’s multiple comparisons test. Logistic regression analyses were conducted to ascertain the independent association of urinary DKD biomarkers with each of the variables, including sex, age, BMI, SBP, serum lipid parameters, UA, Cr, HbA1c, FIB-4 index, HSI, current smoker status, hypertension, dyslipidemia, and duration of T2D. These variables were selected for general glomerular and tubular injury risk factors with reference to previous studies [3,16]. Adjustment for multiple testing was performed using Bonferroni correction.

Receiver operating characteristic (ROC) curve analysis was conducted to determine the optimal cutoff values of serum FIB-4 index in relation to the prevalence of renal tubular injury. The optimal cutoff point selection of FIB-4 index in the context of ROC curve analysis was determined on the basis of the maximum value of the Youden index.

The statistical analyses were performed using Excel (Microsoft Office Excel 16.78.3; Microsoft, Richmond, CA, USA), GraphPad Prism 8.4.3 (GraphPad Software, San Diego, CA, USA), and EZR ver. 1.68 (Saitama Medical Center, Jichi Medical University, Saitama, Japan). In addition, in order to evaluate the statistical power of the sample size in this study, we used G*Power 3 (https://www.psychologie.hhu.de/arbeitsgruppen/allgemeine-psychologie-und-arbeitspsychologie/gpower (accessed on 14 May 2024)), a free software program, to perform power analysis [17]. Statistical significance was defined as *p* < 0.05.

## 3. Results

### 3.1. Clinical Characteristics of the Subjects

The final statistical power based on the obtained sample size for analysis of the results of logistic regression was greater than 99% (Appendix A).

The physical and laboratory characteristics of the subjects in the four groups according to the values of uACR and uL-FABPCR are shown in Table 1. The numbers of subjects were 637 (51%), 248 (20%), 90 (7%), and 268 (22%) for Group N, Group G, Group T, and Group D, respectively. The subjects with elevated uACR levels showed increased SBP regardless of their values of uL-FABPCR. Conversely, the subjects with elevated uL-FABPCR levels were older regardless of their values of uACR. The T group had a significantly higher FIB-4 index than those in the other groups. Additionally, the D group had higher TG, HbA1c and Cr levels, while their eGFR was lower.

In this study, 60.4% of the participants had an FIB-4 index of 1.3 or higher, indicating a risk of liver fibrosis, and 46.9% had an HSI greater than 36, indicating the presence of fatty liver (Appendix A).

### 3.2. Associations of Urinary Biomarkers and Risk Factors across DKD Subtypes

Next, we performed multiple logistic regression analysis to ascertain the independent associations of urinary DKD biomarkers with variables, including male gender, age, BMI, SBP, serum lipid parameters, UA, Cr, HbA1c, FIB-4 index, HSI, current smoking status, presence of hypertension, dyslipidemia, and duration of T2D.

The results of the logistic regression analysis comparing the N group with the G group were as follows: The risk of solitary uACR elevation was associated with BMI (Odds Ratio [OR] = 1.066 (Confidence Interval [CI]: 1.015–1.122), Bonferroni-adjusted *p* = 0.012), current smoking (OR = 2.036 (CI: 1.160–3.564), Bonferroni-adjusted *p* = 0.017), and the presence of hypertension (OR = 1.841 (CI: 1.103–3.127), Bonferroni-adjusted *p* = 0.038). Conversely, in the comparison between the N group and the T group, the risk of solitary uL-FABPCR elevation was associated with Cr level (OR = 7.538 (CI: 1.604–35.51), Bonferroni-adjusted *p* = 0.011), and FIB-4 index (OR = 1.857 (CI: 1.157–3.095), Bonferroni-adjusted *p* = 0.027). BMI (OR = 1.081 (CI: 1.027–1.140), Bonferroni-adjusted *p* = 0.002), HbA1c level (OR = 1.240 (CI: 1.082–1.424), Bonferroni-adjusted *p* < 0.001), Cr level (OR = 23.52 (CI: 8.397–72.70), Bonferroni-adjusted *p* < 0.001), presence of hypertension (OR = 2.134 (CI: 1.224–3.810), Bonferroni-adjusted *p* = 0.009), and duration of T2D (OR = 1.032 (CI: 1.010–1.056), Bonferroni-adjusted *p* = 0.005) emerged as significant risk factors for the D group. Notably, while the presence of hypertension was a distinct specific risk factor for glomerular injury, the FIB-4 index emerged as a specific risk factor for tubular injury in subjects with T2D (Figure 1).

Since pharmacological interventions (including treatments with hypoglycemic agents, antihypertensive agents, and statins) can change the development of DKD, we proceeded to conduct a multiple logistic regression analysis using the established independent variables shown in Figure 1b, along with medications used. Conclusively, the FIB-4 index remained a specific and significant risk factor for the development of tubular injury, regardless of the medications administered to the subjects (Appendix A). Additionally, we conducted an analysis using AST, ALT, and platelets, which are components of the FIB-4 index, as covariates instead of the FIB-4 index. No significant association was found between AST, ALT, platelets, and tubular injury (Appendix A). Additionally, since age is also a component of the FIB-4 index, we conducted an analysis excluding age as a covariate. The results were consistent with those shown in Figure 1b and Appendix A.

To investigate whether not only liver fibrosis but also liver steatosis is involved in tubular injury, we performed logistic regression analysis using HSI, an index of liver steatosis, as a variable instead of the FIB-4 index. The logistic regression analysis revealed that HSI did not show any significant association with the G group, T group, or D group (Figure 2).

Furthermore, we conducted ROC curve analysis to determine the optimal cutoff point of the FIB-4 index for detecting renal tubular injury. The areas under the ROC curves (AUCs) were compared across the entire population and subgroups, specifically individuals with uL-FABPCR ≥ 5. The T group (Figure 3b) had the highest AUC value (95% CI) of 0.623 (0.559–0.686), and the optimal cutoff value of the FIB-4 index for the detection of renal tubular injury was 1.52 with a sensitivity of 59.7% and specificity of 61.1%.

## 4. Discussion

### 4.1. Liver Fibrosis Is Specifically Associated with Tubular Injury in Individuals with T2D

In this study, the FIB-4 index was found to be a significant risk factor for tubular injury in T2D. Conversely, there was no association between HSI and tubular injury. Furthermore, ROC curve analysis to determine the FIB-4 index cutoff value for detecting tubular injury showed the largest AUC in the T group. In addition, while no significant associations were found between platelets or liver transaminase levels, including AST and ALT and tubular injury, an association was observed between the FIB-4 index and tubular injury. A previous report indicated that 56% of type 2 diabetic patients with MASLD have liver transaminase levels within the normal range [18], suggesting that liver transaminase levels may not adequately assess steatohepatitis and fibrosis in patients with T2DM and MASLD. These observations support our results, indicating that liver fibrosis is associated with tubular injury in DKD.

These findings suggest that the subtype of DKD characterized by tubular injury without glomerular injury in T2D is linked to liver fibrosis rather than liver steatosis in MASLD.

Previous cross-sectional and longitudinal studies have shown a significant association between MASLD, which was previously known as nonalcoholic fatty liver disease (NAFLD), and DKD [16,19,20]. However, the precise mechanisms linking MASLD and DKD have remained incompletely understood. In most studies in which the MASLD-DKD association was investigated, DKD was defined by using markers such as albuminuria, proteinuria, and eGFR. Historically, studies on DKD primarily focused on albuminuria as the predominant biomarker reflecting glomerular damage [4]. Nevertheless, cases of DKD lacking significant glomerular alterations have been documented, prompting attention to renal tubular alterations as another critical aspect of DKD [3]. However, there has been no study in which uL-FABP was used as a biomarker of tubular injury in DKD to explore its relationship with MASLD. Therefore, the novelty of this study is the use of uL-FABP to clarify the relationship between tubular injury in DKD and the development of MASLD.

### 4.2. Role of Inflammation in Tubular Injury and Liver Fibrosis in T2D

The association between MASLD and DKD in T2D may be mediated by shared pathogenic metabolic mechanisms underlying both conditions [21]. Insulin resistance plays a pivotal role as a pathological mechanism that is central to both MASLD and T2D. In fact, insulin resistance promotes lipolysis in adipose tissue, resulting in the elevation of plasma-free fatty acid (FFA) concentration [19,22], and hepatic lipid overload fosters the production of lipotoxic lipids, triggering inflammatory responses, mitochondrial dysfunction, and oxidative stress [8], thus contributing to the development and progression of MASLD. This metabolic state extends beyond liver-related inflammation to amplify systemic inflammatory responses, with inflamed adipose tissue secreting various pro-inflammatory cytokines, including tumor necrosis factor-α (TNF-α), interleukin-6 (IL-6), resistin, and monocyte chemoattractant protein-1 (MCP-1) [23]. The transport of pro-inflammatory agents to the kidneys via systemic circulation may serve as initiating or exacerbating factors for kidney diseases. In addition, the elevated levels of pro-inflammatory cytokines induce oxidative stress, leading to apoptosis in tubular epithelial cells (TECs) [24]. Furthermore, a recent animal study by Xuezhu Li et al. revealed that MASLD-induced tubular injury was mediated by mitochondrial dysfunction due to lipid accumulation and vacuolations within TECs in MASH model mice [10]. It has been reported that disturbances in the equilibrium of the gut microbiome due to T2D and obesity are also involved in increased insulin resistance and oxidative stress [25]. Furthermore, the dysbiosis of intestinal flora has been reported to induce dysregulation of intestinal epithelial barrier function and the translocation of bacterial components, leading to hepatic inflammation [26,27] and renal tubulointerstitial injury [26,27]. Thus, there is a possibility that dysbiosis is associated with damage to both the liver and renal tubules in the subjects of our study.

### 4.3. Role of Free Fatty Acid Metabolism in Tubular Injury and Liver Fibrosis in T2D

Additionally, dysregulated expression of enzymes involved in FFA β-oxidation also contributes to lipid accumulation in TECs [28]. It has been reported that peroxisome proliferator-activated receptor alpha (PPARα) expression is downregulated in patients with MASH and negatively correlates with the severity of MASH [29]. The impairment of FFA β-oxidation due to PPARα downregulation causes tubulointerstitial inflammation and fibrosis following ATP depletion, cell death, and intracellular lipid deposition in TECs [30]. It has also been reported that restoring fatty acid metabolism via pharmacological methods protected mice from tubulointerstitial fibrosis [31]. Therefore, normalization of FFA β-oxidation by improving MASLD may prevent the development of both liver fibrosis and tubular injury in DKD.

### 4.4. Role of Advanced Glycation End Products (AGEs) in Tubular Injury and Liver Fibrosis in T2D

AGEs have also been identified as a shared factor between MASLD and tubular injury [32,33,34]. AGEs are complex molecules resulting from nonenzymatic reactions involving glucose or saccharide derivatives and proteins or lipids [35]. In the context of MASLD, AGEs are considered prominent contributors that promote the transition from steatosis to MASH through their interaction with the receptor for AGEs (RAGE). The activation of RAGE, predominantly expressed on Kupffer cells and hepatic stellate cells, serves to advance the progression of MASLD [33]. Within DKD, renal tubular function plays a pivotal role in eliminating free pentosidine, a metabolic byproduct of AGEs [36,37]. Studies have shown that tissue levels of pentosidine are notably elevated in patients with end-stage DKD compared to the levels in patients with diabetes but without renal complications [38]. The interplay of AGEs and RAGE in tubular cells leads to the production of transforming growth factor-beta, a pro-fibrotic factor implicated in the upregulation of type 4 collagen and other fibrotic agents, thereby contributing to tubulointerstitial fibrosis and tubular cell apoptosis [39,40]. Measurement of skin autofluorescence (SAF) serves as a method for detecting pentosidine deposition in the skin. We previously reported that there is a specific and positive correlation between SAF and tubular injury, as indicated by uL-FABP, in individuals with T2D. Interestingly, this correlation is not observed with uACR, suggesting that SAF could potentially serve as a novel marker for identifying tubular injury [41]. Thus, it is thought that AGEs such as pentosidine are pivotal common risk factors that link renal tubular injury and MASLD in individuals with T2D. A schematic diagram showing the hypothesis of interaction between MASLD and renal tubular injury in individuals with T2D based on our results and bibliographical considerations is shown in Appendix A.

### 4.5. Limitations

There are several limitations to the current study. Firstly, the outcomes presented here are not generalizable due to our focus solely on individuals diagnosed with T2D. Secondly, this study did not include imaging examinations to evaluate fibrosis and stiffness of the liver, such as elastography using ultrasound or MRI. An association analysis between tubular injury and severity of liver stiffness assessed using these noninvasive imaging devices would substantiate our hypothesis. Lastly, the inherent limitations of a cross-sectional study impede the elucidation of a causal link between MASLD and the onset of tubular injury, necessitating comprehensive large-scale and longitudinal analyses to address this clinical query definitively.

## 5. Conclusions

Collectively, the findings obtained in this study suggest that the development of liver fibrosis is an inherent risk factor for tubular injury in individuals with T2D. Hence, this study contributes considerable insights into the complex interplay between MASLD and tubular injury in DKD, highlighting the need for comprehensive approaches and targeted interventions addressing metabolic dysregulation and organ-specific complications in individuals with T2D. Continued research in this area is crucial for advancing our understanding and improving clinical outcomes for individuals with these overlapping conditions.

## Figures and Tables

**Figure 1 biomedicines-12-01789-f001:**
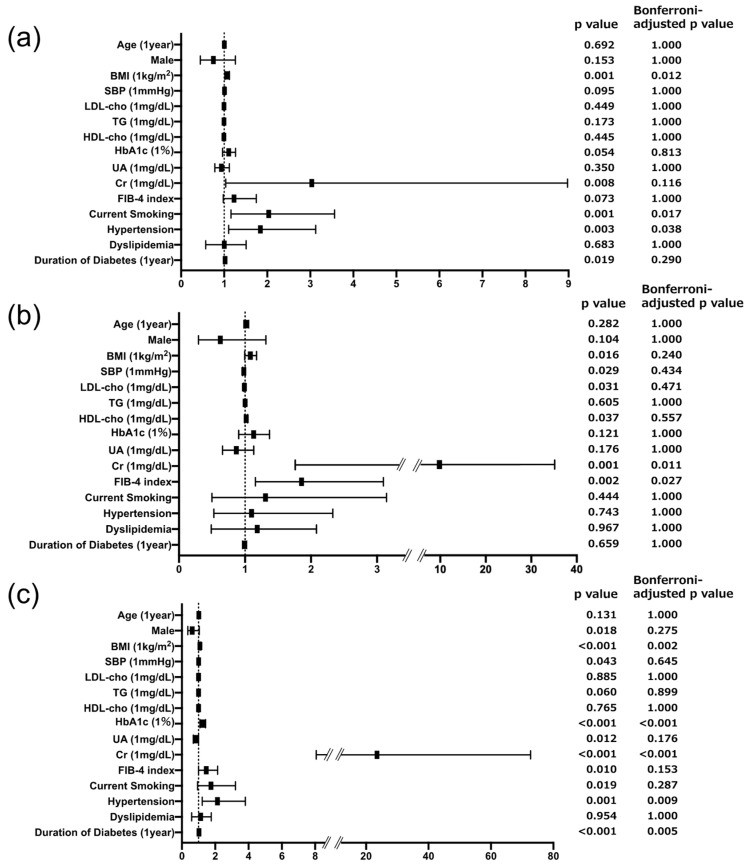
Forest plots showing the results of logistic analysis for independent associations of urinary DKD biomarkers with variables including FIB-4 index. (**a**) Forest plot for prevalence of glomerular specific injury defined as uACR ≥ 30 mg/gCr and uL-FABPCR < 5 μg/gCr. (**b**) Forest plot for prevalence of tubular specific injury defined as uACR < 30 mg/gCr and uL-FABPCR ≥ 5 μg/gCr. (**c**) Forest plot for prevalence of dual injury defined as uACR ≥ 30 mg/gCr and uL-FABPCR ≥ 5 μg/gCr. The dashed line in the graph denotes that a portion of the axis is omitted because one of the data points is too large to display properly in the graph.

**Figure 2 biomedicines-12-01789-f002:**
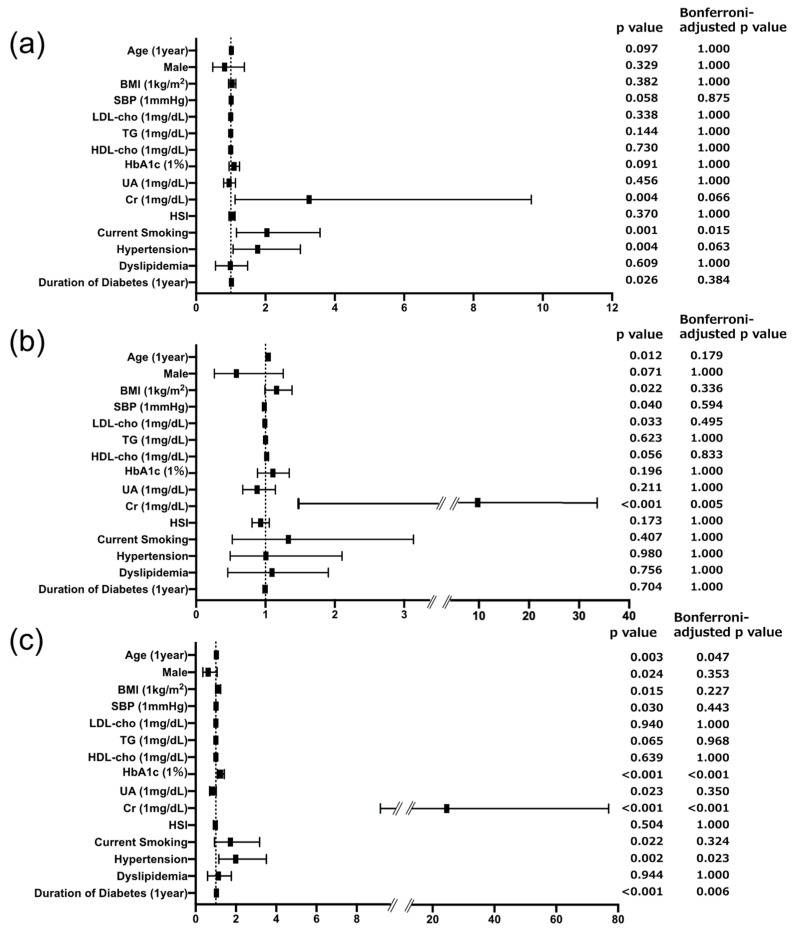
Forest plots showing the results of logistic analysis for independent associations of urinary DKD biomarkers with variables including HSI. (**a**) Forest plot for prevalence of glomerular specific injury defined as uACR ≥ 30 mg/gCr and uL-FABPCR < 5 μg/gCr. (**b**) Forest plot for prevalence of tubular specific injury defined as uACR < 30 mg/gCr and uL-FABPCR ≥ 5 μg/gCr. (**c**) Forest plot for prevalence of dual injury defined as uACR ≥ 30 mg/gCr and uL-FABPCR ≥ 5 μg/gCr. The dashed line in the graph denotes that a portion of the axis is omitted because one of the data points is too large to display properly in the graph.

**Figure 3 biomedicines-12-01789-f003:**
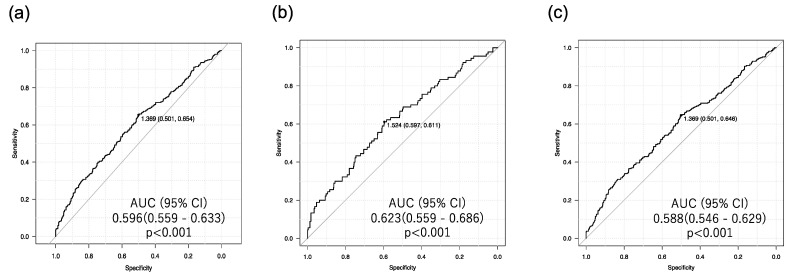
ROC curve analysis to determine the optimal cutoff point of the FIB-4 index for detecting renal tubular injury. (**a**) All subjects (for determination of uL-FABPCR ≥ 5 μg/gCr regardless of uACR). (**b**) T group (for determination of uACR < 30 mg/gCr and uL-FABPCR ≥ 5 μg/gCr). (**c**) D group (for determination of uACR ≥ 30 mg/gCr and uL-FABPCR ≥ 5 μg/gCr).

**Table 1 biomedicines-12-01789-t001:** Clinical characteristics of the subjects.

Group (n)	N (n = 637)	G (n = 248)	T (n = 90)	D (n = 268)	*p* Value
Age (years)	68.0 (58.4, 74.0)	70.0 (60.0, 76.0)	72.0 (62.2, 78.0) *	73.0 (62.0, 78.5) *	<0.001
Male (n, (%))	349 (54.8)	142 (57.3)	49 (54.4)	159 (59.3)	0.614
BMI (kg/m^2^)	24.5 (21.9, 27.3)	25.0 (22.8, 28.0)	23.7 (22.4, 26.7)	25.2 (22.4, 28.2) *	0.005
SBP (mmHg)	130 ± 16	135 ± 18 *	127 ± 17	136 ± 17 *^‡^	<0.001
LDL-cho (mg/dL)	99 (82, 120)	97 (76, 116)	88 (74, 106)	93 (77, 115)	0.003
TG (mg/dL)	110 (78, 159)	122(84, 161)	110 (81, 157)	132 (85, 197) *	0.009
HDL-cho (mg/dL)	53 (45, 63)	52 (44, 63)	55 (47, 65)	53 (43, 62)	0.140
Casual PG (mg/dL)	131 (112, 165)	137 (118, 175)	140 (111, 186)	147 (121, 193) *^†^	<0.001
HbA1c (%)	6.8 (6.3, 7.4)	6.9 (6.5, 7.4)	6.8 (6.4, 7.3)	7.1 (6.6, 8.0) *^†‡^	<0.001
UA (mg/dL)	5.0 (4.1, 5.9)	5.0 (4.2, 5.8)	4.8 (4.1, 5.8)	5.2 (4.3, 5.9)	0.309
Cr (mg/dL)	0.73 (0.60, 0.87)	0.75 (0.61, 0.93)	0.80 (0.59, 1.09)	0.86 (0.69, 1.15) *^†^	<0.001
eGFR (ml/min/1.73 m^2^)	74.8 ± 18.0	72.4 ± 21.5	70.1 ± 26.4	61.6 ± 22.8 *^†^	<0.001
Albumin (g/dL)	4.2 ± 0.4	4.2 ± 0.4	4.1 ± 0.4	4.2 ± 0.5	0.069
AST (U/L)	20 (17, 25)	21 (18, 28)	22 (18, 28)	22 (18, 29) *	0.009
ALT (U/L)	21 (15, 29)	21 (16, 30)	19 (14, 31)	20 (14, 33)	0.568
Platelets (10^9^/L)	216 (189, 256)	222 (188, 261)	208 (179, 260)	220 (182, 266)	0.671
uACR (mg/gCr)	9.8 (6.0, 15.6)	58.8 (41.3, 123.3) *^‡^	14.7 (8.6, 20.3)	209.5 (78.0, 550.5) *^†‡^	<0.001
uL-FABPCR (μg/gCr)	2.0 (1.3, 3.0)	2.6 (1.9, 3.5) *	6.7 (5.5, 9.0) *^†^	13.1 (7.6, 21.7) *^†^	<0.001
FIB-4 index	1.4 (1.0, 1.9)	1.5 (1.1, 2.0)	1.7 (1.3, 2.3) *	1.6 (1.1, 2.2)	<0.001
HSI	35.6 (32.1, 39.3)	35.8 (32.6, 41.0)	34.5 (31.1, 38.4)	36.1 (32.0, 40.9) *	0.048
Current Smoking (n, (%))	101 (15.9)	54 (21.8)	14 (15.6)	51 (19.0)	0.181
Hypertension (n, (%))	383 (60.1)	194 (78.2)	59 (65.6)	225 (84.0)	<0.001
Dyslipidemia (n, (%))	459 (72.1)	184 (74.2)	63 (70.0)	215 (80.2)	0.053
Duration of T2D (years)	9 (3, 15)	11 (5, 18) *	11 (5, 17)	13 (7, 22) *	<0.001
ARB or ACEi (n, (%))	222 (34.9)	136 (54.8) *	43 (47.8)	140 (52.2) *	<0.001
CCB (n, (%))	193 (30.3)	121 (48.8) *	35 (38.9)	165 (61.6) *^†‡^	<0.001
β blocker (n, (%))	46 (7.2)	16 (6.5)	6 (6.7)	24 (9.0)	0.730
MR antagonist (n, (%))	15 (2.4)	9 (3.6)	1 (1.1)	6 (2.2)	0.620
Statin (n, (%))	305 (47.9)	135 (54.4)	46 (51.1)	133 (49.6)	0.595
Ezetimibe (n, (%))	51 (8.0)	14 (5.6)	6 (6.7)	33 (12.3)	0.050
Other Hypolipidemics (n, (%))	16 (2.5)	13 (5.2)	4 (4.4)	15 (5.6)	0.061
Antiplatelet (n, (%))	67 (10.5)	33 (13.3) *	8 (8.9)	48 (17.9) *	0.017
SU or Glinide (n, (%))	88 (13.8)	49 (19.8)	22 (24.4)	77 (28.7) *^†^	<0.001
Metformin (n, (%))	313 (49.1)	133 (53.6)	40 (44.4)	123 (45.9)	0.273
Imeglimin (n, (%))	17 (2.7)	5 (2.0)	1 (1.1)	8 (3.0)	0.833
DPP-4i (n, (%))	331 (52.0)	137 (55.2)	58 (64.4)	132 (49.3)	0.071
SGLT2i (n, (%))	215 (33.8)	106 (42.7)	34 (37.8)	137 (51.1) *^†‡^	<0.001
αGI (n, (%))	79 (12.4)	43 (17.3)	19 (21.1)	41 (15.3)	0.065
Pioglitazone (n, (%))	17 (2.7)	9 (3.6)	3 (3.3)	17 (6.3)	0.075
Insulin (n, (%))	139 (21.8)	57 (23.0)	21 (23.3)	76 (28.4)	0.214
GLP-1RA (n, (%))	101 (15.9)	58 (23.4) *	19 (21.1)	75 (28.0) *	<0.001

* *p* < 0.05 vs. N group, ^†^ *p* < 0.05 vs. G group, ^‡^ *p* < 0.05 vs. T group (one-way ANOVA followed by Dunn’s multiple comparisons test). The values are presented as means ± SD or medians (Q1, Q3). Abbreviations: BMI: body mass index; SBP: systolic blood pressure; TG: triglycerides; HDL-C: high-density lipoprotein cholesterol; LDL-C: low-density lipoprotein cholesterol; PG: plasma glucose; HbA1c: hemoglobin A1c; UA: uric acid; Cr: creatinine; eGFR: estimated glomerular filtration rate; uACR: urinary albumin-to-creatinine ratio; uL-FABPCR: urinary liver-type fatty acid-binding protein-to-creatinine ratio; FIB-4; fibrosis-4; HSI; hepatic steatosis index; ARB: angiotensin II receptor blocker; ACEi: angiotensin-converting enzyme inhibitor; CCB: calcium channel blocker; MR: mineral corticoid receptor; SU: sulfonylurea; DPP-4i: dipeptidyl peptidase-4 inhibitor; SGLT2i: sodium–glucose cotransporter 2 inhibitor; αGI: alpha-glucosidase inhibitor; GLP-1RA: glucagon-like peptide-1 receptor agonist.

## Data Availability

The datasets generated in the present study are available from the corresponding author upon reasonable request.

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
