# Peer review of "Development of Liver Fibrosis Represented by the Fibrosis-4 Index Is a Specific Risk Factor for Tubular Injury in Individuals with Type 2 Diabetes"

_biomedicines, 2024, doi:10.3390/biomedicines12081789_

Round 1

Reviewer 1 Report

Comments and Suggestions for Authors

The manuscript submitted by Hara et al., titled: "Development of Liver Fibrosis Represented by The Fibrosis-4 Index Is a Specific Risk Factor for Tubular Injury in Individuals with Type 2 Diabetes" is a human study aiming to investigate how a particular index (F4I) used to assess liver fibrosis can function as a specific risk factor (predictive measure) for tubular injury in people with T2DM.

This is a well written paper with a good structure and logical organization. The reviewer would like to offer a few points for consideration by the authors:

1. Consider providing a more detailed account on the inclusion and exclusion criteria for the participants.

2.  How was the the sample size for the population determined?

3. Did the authors consider and how did they normalize for confounding factors (sex, smoking, medications, diet, physical activity etc)?

4. BMI does not have units. 

5. Did the authors consider body composition?

6. It would be interesting to consider in the discussion the relationship between microbiome and diabetes. An interesting review article that may help to that end is the following: Sikalidis, A.K.; Maykish, A. The Gut Microbiome and Type 2 Diabetes Mellitus: Discussing A Complex Relationship. Biomedicines 20208, 8. https://doi.org/10.3390/biomedicines8010008.

Good job overall.

Author Response

Reviewer 1

Comments and Suggestions for Authors

The manuscript submitted by Hara et al., titled: "Development of Liver Fibrosis Represented by The Fibrosis-4 Index Is a Specific Risk Factor for Tubular Injury in Individuals with Type 2 Diabetes" is a human study aiming to investigate how a particular index (F4I) used to assess liver fibrosis can function as a specific risk factor (predictive measure) for tubular injury in people with T2DM.

This is a well written paper with a good structure and logical organization. The reviewer would like to offer a few points for consideration by the authors:

  1. Consider providing a more detailed account on the inclusion and exclusion criteria for the participants.

Response: In accordance with the reviewer’s comment, we added a more detailed account on the inclusion and exclusion criteria for the participants. We revised sentences about this issue in lines 79 to 88 of the revised manuscript.

  1. How was the sample size for the population determined?

Response: As this study was a retrospective clinical study, no preliminary sample size calculation was conducted. However, we performed a post hoc power analysis. The final power based on the obtained sample size for analysis of the results of logistic regression was greater than 98%. We added sentences about this issue to the methods and results sections in lines 156 to 160 and 164 to 165 of the revised manuscript and added Figure S1(a), (b) and (c).

  1. Did the authors consider and how did they normalize for confounding factors (sex, smoking, medications, diet, physical activity etc)?

Response: To determine optimal variables for logistic regression analysis, we selected general glomerular and tubular injury risk factors with reference to previous studies. (R. Retnakaran et al, Diabetes 2006; 55(6):1832-1839, G. Targher, et al, Diabetologia 2008;51(3):444-450). We added sentences about this issue in lines 146 to 149 of the revised manuscript. Unfortunately, we were not able to obtain data for diet and physical activity.

  1. BMI does not have units. 

Response: In many research articles, BMI is commonly expressed in units with kg/m2, so we chose to retain this convention.

  1. Did the authors consider body composition?

Response:In this study, we did not have any data on body composition of the subjects. We appreciate the reviewer's suggestion, and we will consider it in our future research.

  1. It would be interesting to consider in the discussion the relationship between microbiome and diabetes. An interesting review article that may help to that end is the following: Sikalidis, A.K.; Maykish, A. The Gut Microbiome and Type 2 Diabetes Mellitus: Discussing A Complex Relationship. Biomedicines2020, 8, 8. https://doi.org/10.3390/biomedicines8010008.

Response: Thank you for your valuable comment. According to the suggestion by the reviewer, we added discussion of the relationship between microbiome and diabetes in lines 280 to 287 of the revised manuscript.

Reviewer 2 Report

Comments and Suggestions for Authors

I have read and analyzed the manuscript from Hara and coauthors. In my opinion, the manuscript is well-writen but the conclusions of the manuscript are ambitious. My remarks are:

1.Authors should clarified the manufacturers and catalogue numbes for all used kits in study.

2.Table 1. I recommend to mark all significant differencies additionally.

3.The results of the study do not confirm the last scheme in Figure 3.

Moreover, the amount of data in this manuscript is too small, I can highly recommend enlarge the dataset for publication. In my opinion, the manuscript needs extensive review.

Author Response

Reviewer 2

Comments and Suggestions for Authors

I have read and analyzed the manuscript from Hara and coauthors. In my opinion, the manuscript is well-writen but the conclusions of the manuscript are ambitious. My remarks are:

  1. Authors should clarified the manufacturers and catalogue numbes for all used kits in study.

Response: Urinary levels of ACR were measured using TIA-ALBG (No. 20600AMZ01334000, Serotec, Hokkaido, Japan). Urinary levels of L-FABP were measured using Lumipulse Presto L-FABP (No. 298213, Fujirebio, Tokyo, Japan). We added this information to the methods section in lines 111 to 114 of the revised manuscript.

  1. Table 1. I recommend to mark all significant differencies additionally.

Response: In accordance with the reviewer’s comment, we added markers for all significant differences to Table 1 of the revised manuscript.

3.The results of the study do not confirm the last scheme in Figure 3.

Moreover, the amount of data in this manuscript is too small, I can highly recommend enlarge the dataset for publication. In my opinion, the manuscript needs extensive review.

Response: In response to the reviewer's comments, we have decided not to treat Figure 3 as a main text figure, since it represents a hypothesis, including our results. We have therefore moved it to the Supplemental Data (Figure S4 of the revised manuscript), with the source indicated for the previously identified aspects of the literature discussion. The sentences of the discussion were partially changed accordingly (in lines 322 to 324 of the revised manuscript).

As noted in the Limitations section, since this study was a cross-sectional study, we were unable to determine causal relationships. We appreciate the reviewer's valid point and acknowledge that a large-scale longitudinal study is necessary to clarify the causal relationship. However, the power analysis confirmed that the sample size in our study is sufficient for the logistic analysis, as detailed in our response to Reviewer 1.

Reviewer 3 Report

Comments and Suggestions for Authors

The authors present data from a cross-sectional cohort study, investigating predictors for specific renofunctional disturbances.

The overall rationale is not clear. Why would liver fibrosis (and not plain NAFLD or any other co-morbidity) induce tubular (or another type of) renal defect? What is the putative mechanism?

Introduction: see above

Methods: Mainly fine.

FIB-4 contains age as parameter. Can you outrule, that age is the actual important predictor instead of liver enzymes? Among the FIB-4 components, age seems to be one which discriminates the four reno-functional strata in the strongest fashion. Has the Fatty Liver Index (or the HSI) or the TyG index as measure for insulin resistance a similar, weaker or stronger predictive capability?

Fib-4 values in your cohort are mostly low; fibrosis is highly probable at values > 3,25, for elderly patients even just probable at values > 2,0. Only few patients in your cohort fulfill these criteria.

Results:
Please use adequate decimals for continuous parameters; i.e. only as many decimals as raw data precision allows for (e.g. BMI with one, LDL and RR without).

For the analysis in Fig. 1, correction for multiple testing is required.

Discussion: Can only be done after major revision.

Comments on the Quality of English Language

minor changes needed

Author Response

Reviewer 3

comments and Suggestions for Authors

The authors present data from a cross-sectional cohort study, investigating predictors for specific renofunctional disturbances.

The overall rationale is not clear.

Why would liver fibrosis (and not plain NAFLD or any other co-morbidity) induce tubular (or another type of) renal defect? What is the putative mechanism?

Response: Since this study was a cross-sectional study, the causal relationship between tubular injury and liver fibrosis as expressed by the FIB-4 index has not been determined. However, based on our current findings and previous research, we hypothesize that the mechanisms involved in the development of liver fibrosis, such as chronic inflammation, oxidative stress, free fatty oxidation and advanced glycation end-products, are also involved in tubular injury, rather than just fatty liver.

Furthermore, an additional logistic regression analysis using Hepatic Steatosis Index (HSI) (Jeong-Hoon Lee et.al., Dig Liver Dis. 2010 Jul;42(7):503-8), a marker of the severity of liver steatosis, was conducted in accordance with the suggestion of the reviewer. The logistic regression analysis revealed that HSI did not show a significant association with the glomerular injury alone group, tubular injury alone group, or both injury group. These findings further support our hypothesis that the risk of tubular injury may be related to the development of liver fibrosis rather than being solely attributed to liver steatosis.

Results and discussion of logistics analysis using HSI as a variable were added in lines 217 to 221 and indicated as Figure 2 of the revised manuscript. With these additions, the research background section has been reconsidered as described in the revised version (in lines 61 to 67, 73 to 75 of the revised manuscript.)

Introduction: see above

Methods: Mainly fine.

FIB-4 contains age as parameter. Can you out rule, that age is the actual important predictor instead of liver enzymes? Among the FIB-4 components, age seems to be one which discriminates the four reno-functional strata in the strongest fashion.

Response: In the logistic analysis for which results are shown in Figure 1, age was included as a covariate, and the FIB-4 index remained significant as a tubular injury- specific risk factor even after adjusting for age.

Has the Fatty Liver Index (or the HSI) or the TyG index as measure for insulin resistance a similar, weaker or stronger predictive capability?

Response: Unfortunately, we were unable to calculate the Fatty Liver Index in our study due to a lack of abdominal circumference data. The TyG index was also not able to be shown due to casual but not fasting plasma glucose data in this study. However, we conducted an additional analysis using HSI, and the results are described above.

Fib-4 values in your cohort are mostly low; fibrosis is highly probable at values > 3,25, for elderly patients even just probable at values > 2,0. Only few patients in your cohort fulfill these criteria.

Response: According to a report in which the cutoff values of the FIB-4 index are presented (Amy G Shah et al., Clin Gastroenterol Hepatol. 2009 Oct;7(10):1104-12), an FIB-4 index < 1.3 is categorized as low risk, an FIB-4 index from 1.3 to less than 2.67 is categorized as a high risk of fibrosis, while an FIB-4 index ≥ 2.67 is categorized as cirrhosis. The clinical practice guidelines established by the European Association for the Study of the Liver (The EASL Clinical Practice Guidelines on non-invasive tests for assessing liver disease severity and prognosis - 2021 update) recommend consulting a hepatologist if the FIB-4 index is 1.3 or higher because of the high risk of liver fibrosis. In our study, approximately 60% of all cases had an FIB-4 index of 1.3 or higher (Histograms were added to Figure S2.). In addition, the median FIB-4 index of the T group was around 2.0, which was significantly higher than the median values in the other groups, and there was a greater distribution of high risk of liver fibrosis in the T group. These results support our conclusion.

Results:
Please use adequate decimals for continuous parameters; i.e. only as many decimals as raw data precision allows for (e.g. BMI with one, LDL and RR without).

Response: In accordance with the reviewer's comment, we modified the parameters of LDL-C, TG, HDL-C, casual glucose, AST, ALT, platelets, and duration of diabetes without a decimal point in Table 1 of the revised manuscript.

For the analysis in Fig. 1, correction for multiple testing is required.

Response: In accordance with the reviewer's comment, we appended Bonferroni-adjusted p values in Figure 1, and significant differences were determined by the re-analyzed p values. Even after the adjustment, the FIB-4 index remained significantly correlated with p=0.012 in the tubular injury-specific group. The sentences of the methods and results were partially changed accordingly (in lines 148 to 149 and 193 to 202 of the revised manuscript).

Reviewer 4 Report

Comments and Suggestions for Authors

Manuscript ID: biomedicines-2991773

Title: "Development of Liver Fibrosis Represented by The Fibrosis-4 2 Index Is a Specific Risk Factor for Tubular Injury in Individuals 3 with Type 2 Diabetes"

Authors: Tomoyo Hara et al.

The authors in the present study explored risk factors for glomerular and tubular injury in individuals with type 2 diabetes (T2D). They categorized individuals into four groups based on urinary biomarkers and identified distinct risk factors for each group. Analyses revealed that BMI, hypertension, and duration of T2D were common risk factors for glomerular injury, while BMI, creatinine levels, and the Fibrosis-4 index were specifically associated with tubular injury. The study hypothesis is interesting; however, the following comments should be considered:

Comments:

1.     The authors state, " While BMI is a common risk factor for all types of renal injuries, the FIB-4 index was a specific contributor for the prevalence of tubular injury." However, the results presented suggest that creatinine is also a relevant factor. Please also consider this.

2.     Please ensure that appropriate references are included for each statement made in the introduction.

3.     In Table 1, could you please include information regarding the participants' sex?

4.     Have the authors considered assessing insulin resistance by calculating the relevant indices and including these in their analyses?

5.     In the discussion section, adding a separate paragraph before the limitations that briefly outlines the clinical implications of the findings would be beneficial for readers.

Author Response

Reviewer 4

Comments and Suggestions for Authors 

The authors in the present study explored risk factors for glomerular and tubular injury in individuals with type 2 diabetes (T2D). They categorized individuals into four groups based on urinary biomarkers and identified distinct risk factors for each group. Analyses revealed that BMI, hypertension, and duration of T2D were common risk factors for glomerular injury, while BMI, creatinine levels, and the Fibrosis-4 index were specifically associated with tubular injury. The study hypothesis is interesting; however, the following comments should be considered:

Comments:

  1. The authors state, " While BMI is a common risk factor for all types of renal injuries, the FIB-4 index was a specific contributor for the prevalence of tubular injury." However, the results presented suggest that creatinine is also a relevant factor. Please also consider this.

Response: Serum creatinine was not significantly associated with the G group but emerged as a significant risk factor in groups T and D. Therefore, we stated that only BMI is a common risk factor for all types of renal injuries in the original version. Based on these findings, we consider the possibility that the influence of creatinine on glomerular injury becomes apparent after the occurrence of tubular injury.

  1. Please ensure that appropriate references are included for each statement made in the introduction.

Response: In accordance with the reviewer's comments, I have reviewed and cited the references in the introduction section (in lines 48, 54, 57, and 59 of the revised manuscript). Thank you for your advice.

  1. In Table 1, could you please include information regarding the participants' sex?

Response: In accordance with the reviewer’s comment, we added information regarding the participants' sex in Table 1.

  1. Have the authors considered assessing insulin resistance by calculating the relevant indices and including these in their analyses?

Response: The issue that the reviewer raised is crucial. However, our study relied on casual blood sampling, which resulted in the inability to calculate indices of insulin resistance such as HOMA-R or the TyG index. We will investigate the association between insulin resistance and tubular injury in a future study.

  1. In the discussion section, adding a separate paragraph before the limitations that briefly outlines the clinical implications of the findings would be beneficial for readers.

Response: In accordance with the reviewer’s comment, we added briefly outlines in the discussion section of the revised manuscript.

Round 2

Reviewer 1 Report

Comments and Suggestions for Authors

The authors made a reasonable effort to address reviewer's comments.

Author Response

Thank you for your positive feedback. We appreciate your time and effort in reviewing our manuscript.

We are glad our revisions have addressed your comments satisfactorily.

Thank you again for your support.

Reviewer 2 Report

Comments and Suggestions for Authors

Many thanks to the authors for the resolving of raised questions. However, I still have a doubts in sufficient manuscript volume. This question will be leave to the discretion of Editor.

Author Response

Reviewer 2

Many thanks to the authors for the resolving of raised questions. However, I still have a doubt in sufficient manuscript volume. This question will be leave to the discretion of Editor.

Response: We acknowledge the reviewer's concern about the sufficiency of the data volume. To address this, we have increased our sample size from 864 to 1,243. This addition ensures a more robust dataset.

A power analysis was conducted to determine the adequacy of the sample size. The increased sample size now provides a power of 99%, which is well within the acceptable range for ensuring the reliability of results. With the increased sample size, we have reanalyzed the data and found that the results remain consistent with our initial findings, further validating the robustness of our conclusions.

We believe that these measures adequately address the reviewer's concerns and demonstrate the robustness of our study.

Reviewer 3 Report

Comments and Suggestions for Authors

The authors have revised their manuscript in accordance to the reviewer's suggestions.

Few points should be improved further:

1) Decimals should be shortened also for blood pressure.

2) The rationale is still not clear. Why would liver fibrosis specifically promote tubular - but not glomerular - defects? What do you suppose is the highly specific mechanism behind that? There must have been a reason for assuming, that glomerular damage is independent from liver fibrosis, BEFORE conducting your study.

3) As the analysis was adjusted for age, but neither platelets, nor BMI nor transaminase levels strongly differ between the four groups, it is surprising, that FIB-4 should have a significant association. Also, including age in the FIB-4, but then adjusting for age, when analysing FIB-4, does not seem like a proper way of analysis. You could just analyse the association with AST, ALT, AST/ALT ratio or platelets, instead.

4) Bonferroni correction applies a factor to all p values of a set of analysis, the factor being the number of individual analyses within each set, not the number of sets. For Fig. 1A-C and 2A-C, this factor is not 3, but 15.

5) Is it possible, that certain medication leads to lower platelet levels, thus pretending the existence of liver fibrosis by increasing FIB-4 irrespective of actual fibrosis?

6) SGLT-2 inhibitors may be renoprotective by reducing hyperfiltration, thus lowering mean eGFR (even though it is only happening in patients with supranormal values). This medication needs to be accounted for.

7) The ROC analyses lack p values.

8) If fasting glucose was not available, triglycerides are not fasted either. They hardly qualify as predictive parameter then.

Author Response

Reviewer 3

The authors have revised their manuscript in accordance with the reviewer's suggestions. Few points should be improved further:

1) Decimals should be shortened also for blood pressure.

Response: In accordance with the reviewer's comment, we modified the blood pressure values to exclude decimal points in Table 1 of the revised manuscript.

2) The rationale is still not clear. Why would liver fibrosis specifically promote tubular - but not glomerular - defects? What do you suppose is the highly specific mechanism behind that? There must have been a reason for assuming, that glomerular damage is independent from liver fibrosis, BEFORE conducting your study.

Response: Thank you for your insightful comments. As stated in the limitations, this study was a cross-sectional study and the causal relationship between tubular damage and liver fibrosis could therefore not be elucidated. To clarify the causal relationship, large-scale longitudinal studies and basic research are required, and we plan to conduct these in the future. There have been reports indicating an association between liver fibrosis and DKD (Sci Rep. 2021 Jun 3;11(1):11753). However, it remains unclear whether this association pertains to the progression of all types of DKD or is specific to certain types. Considering that some basic research reports suggest the possibility of a common pathological mechanism underlying both liver fibrosis and renal tubular fibrosis (Kidney Int. 2024 Mar;105(3):540-561), we hypothesized that liver fibrosis might be more strongly related to the type of DKD that presents with tubular injury. Therefore, we conducted this study to clarify how liver fibrosis in MASLD is associated with each type of DKD.

To enhance the understanding of our research background, we have revised the introduction section and included additional references in the revised version (in lines 73 to 79 of the revised manuscript).

In addition, as for factors other than Cr value that caused glomerular injury, the influence of current smoking and presence of hypertension was overwhelmingly greater than that of liver fibrosis, so the results of this study may have weakened the correlation between the FIB-4 index and albuminuria.

3) As the analysis was adjusted for age, but neither platelets, nor BMI nor transaminase levels strongly differ between the four groups, it is surprising, that FIB-4 should have a significant association. Also, including age in the FIB-4, but then adjusting for age, when analyzing FIB-4, does not seem like a proper way of analysis. You could just analyze the association with AST, ALT, AST/ALT ratio or platelets, instead.

Response: In accordance with the reviewer’s comment, we conducted an analysis using AST, ALT, and platelets as covariates instead of the FIB-4 index (Figure S4 (a)) and we found no significant association between platelets or liver transaminase levels including AST and ALT and tubular injury. A previous report indicated that 56% of type 2 diabetic patients with MASLD have liver transaminase levels within the normal range (J Clin Endocrinol Metab. 2015 Jun;100(6):2231-8), suggesting that liver transaminase levels may not adequately assess steatohepatitis and fibrosis in patients with T2DM and MASLD. In this study, while no significant associations of AST, ALT and platelets with tubular injury were found, a correlation was observed between FIB-4 index and tubular injury. These findings support our results indicating that liver fibrosis is associated with tubular injury in DKD. Additionally, in response to the concern that using age as a covariate might be inappropriate due to its inclusion in the FIB-4 index components, we performed logistic regression analysis with a model in which age was excluded from the covariates (Figure S4 (b)). The results were consistent with those obtained when age was included. These findings have been added to the revised manuscript and supplemental data (in lines 219 to 224, 261 to 267 of the revised manuscript and Figure S4).

4) Bonferroni correction applies a factor to all p values of a set of analysis, the factor being the number of individual analyses within each set, not the number of sets. For Fig. 1A-C and 2A-C, this factor is not 3, but 15.

Response: In accordance with the reviewers' comments, we increased the sample size sufficiently to conduct Bonferroni correction using the 15 covariates. After Bonferroni correction with the 15 covariates, the FIB-4 index persisted as a significant risk factor for tubular injury within the T group. Furthermore, we subjected the logistic regression analysis using HSI as a covariate to Bonferroni correction; nevertheless, no discernible association between HSI and DKD was observed across any groups. These findings are delineated in the revised Figures 1 and 2.

5) Is it possible, that certain medication leads to lower platelet levels, thus pretending the existence of liver fibrosis by increasing FIB-4 irrespective of actual fibrosis?

Response: In accordance with the reviewers' comments regarding the potential impact of medications on platelet count, we conducted logistic regression analysis using the established independent variables shown in Figure 1(b) along with medications used. As shown in Figure S3, regardless of pharmacological interventions, the FIB-4 index emerged as a significant risk factor for tubular injury in DKD.

6) SGLT-2 inhibitors may be renoprotective by reducing hyperfiltration, thus lowering mean eGFR (even though it is only happening in patients with supranormal values). This medication needs to be accounted for.

Response: As the reviewer pointed out, since pharmacological interventions, including treatment with SGLT2 inhibitors, can alter the development of DKD, we conducted a multiple logistic regression analysis using the established independent variables shown in Figure 1(b) along with medications used, and the results are shown in Figure S3. The analysis revealed that the FIB-4 index remained a specific and significant risk factor for the development of tubular injury regardless of pharmacological interventions.

7) The ROC analyses lack p values.

Response: In accordance with the reviewer's comment, we added p values of the ROC curve analysis in Figure 3 of the revised manuscript.

8) If fasting glucose was not available, triglycerides are not fasted either. They hardly qualify as predictive parameter then.

Response: Traditionally, triglycerides were assessed through fasting blood samples; however, there are reports suggesting that non-fasting measurements might be even more predictive of cardiovascular events (Atherosclerosis, 2014 Nov;237(1):361-8). Consensus statements from the European Atherosclerosis Society (EAS) and European Federation of Clinical Chemistry and Laboratory Medicine (EFLM) also advocate the use of non-fasting triglyceride values in evaluation (Eur Heart J, 2016 Jul 1;37(25):1944-58). Considering these points, non-fasting triglyceride levels are also appropriate as predictive factors.

Reviewer 4 Report

Comments and Suggestions for Authors

Manuscript ID: biomedicines-2991773 (Revised version)

Title: "Development of Liver Fibrosis Represented by the Fibrosis-4 2 Index Is a Specific Risk Factor for Tubular Injury in Individuals with Type 2 Diabetes"

Authors: Tomoyo Hara et al.

There are no further comments.

Author Response

Thank you for your thorough evaluation and feedback on our manuscript.

We are pleased to hear that there are no further comments. Your insights have significantly improved the quality of our paper, and we are grateful for your contribution.

Thank you once again for your support.

Round 3

Reviewer 3 Report

Comments and Suggestions for Authors

The authors have answered all questions and revised all necessary sections in accordance to the reviewer's suggestions. I consider the paper ready for acceptance.